# Sharp Spectral Rates for Koopman Operator Learning

**Vladimir R. Kostic**
Istituto Italiano di Tecnologia
University of Novi Sad
`vladimir.kostic@iit.it`

**Karim Lounici**
CMAP-Ecole Polytechnique
`karim.lounici@polytechnique.edu`

**Pietro Novelli**
Istituto Italiano di Tecnologia
`pietro.novelli@iit.it`

**Massimiliano Pontil**
Istituto Italiano di Tecnologia
University College London
`massimiliano.pontil@iit.it`

## Abstract

Nonlinear dynamical systems can be handily described by the associated Koopman operator, whose action evolves every observable of the system forward in time. Learning the Koopman operator and its spectral decomposition from data is enabled by a number of algorithms. In this work we present for the first time non-asymptotic learning bounds for the Koopman eigenvalues and eigenfunctions. We focus on time-reversal-invariant stochastic dynamical systems, including the important example of Langevin dynamics. We analyze two popular estimators: Extended Dynamic Mode Decomposition (EDMD) and Reduced Rank Regression (RRR). Our results critically hinge on novel minimax estimation bounds for the operator norm error, that may be of independent interest. Our spectral learning bounds are driven by the simultaneous control of the operator norm error and a novel metric distortion functional of the estimated eigenfunctions. The bounds indicates that both EDMD and RRR have similar variance, but EDMD suffers from a larger bias which might be detrimental to its learning rate. Our results shed new light on the emergence of spurious eigenvalues, an issue which is well known empirically. Numerical experiments illustrate the implications of the bounds in practice.

## 1 Introduction

Recently, researchers have emphasized the utmost importance of developing physically-informed machine learning models that prioritize interpretability and foster physical insight and intuition, see for example [22] and references therein. One technique highlighted in these works is the Koopman operator regression framework to learn and interpret nonlinear dynamical systems see, e.g. [8, 26] and references therein. A key component of this approach is the Koopman Mode Decomposition (KMD), which decomposes complex dynamical systems into simpler, coherent structures. When ordinary least squares are used to learn Koopman operator from data, estimated KMD is known as the Dynamic Mode Decomposition (DMD) [36]. Koopman operator estimators and their modal decomposition find many applications, including fluid dynamics, molecular kinetics and robotics [7, 19].

The Koopman operator returns the expected value of observables of the system in the future given the present, and one relies on estimators of this operator to in turn estimate its spectral decomposition that leads to the estimation of KMD. Our goal is to study the statistical properties of the eigenvalues and eigenfunctions of the Koopman operator estimators via two mainstream algorithms: Principal Component Regression (PCR) and Reduced Rank Regression (RRR) studied in [21, 24]. PCR encompasses as particular cases the popular Extended Dynamic Mode Decomposition (EDMD), which is the de-facto estimator in the data-driven dynamical system literature [see 26, 45, and references

37th Conference on Neural Information Processing Systems (NeurIPS 2023).

therein]. Both PCR and RRR are kernel-based algorithms that, given a dataset of observations of the dynamical system, implement a strategy to approximate the action of the Koopman operator on a reproducing kernel Hilbert space (RKHS) [3, 38].

We present for the first time non-asymptotic learning bounds on the distance between the Koopman eigenvalues and eigenfunctions and those estimated by either PCR or RRR. We show that the eigenvalues produced by such algorithms are *biased* estimators of the true Koopman eigenvalues, with PCR incurring larger bias. Our results critically hinge on novel estimation bounds for the operator norm error, that may be of independent interest, leading to minimax optimal bounds for finite-rank Koopman operators. Moreover, we introduce the novel notion of metric distortion, which characterize how the norm of eigenfunctions vary when moving from the RKHS in which learning takes place to the underlying ambient space where the Koopman operator is properly defined. We show that both the operator norm error and metric distortion are needed in order to estimate the operator spectra and our bounds can be used to explain the well-known spuriousness phenomena in eigenvalue estimation [14], namely, the scenario in which the estimated eigenvalues are not related to the true ones, despite small operator norm error.

**Contributions and Organization.** We make the following contributions: **i)** We introduce the notion of metric distortion and show that it has to be used alongside the operator norm error to derive Koopman spectra estimation error bounds (Theorem 1); **ii)** We establish the first sharp estimation bound for the operator norm error (Theorem 2); **iii)** We establish spectral learning rates (Thms. 3 and 4) for both PCR and RRR; **iv)** We propose how to use the results entailed by Theorem 4 to detect the presence of spurious eigenvalues from data.

The paper is organized as follows. In Section 2 we recall the notion of Koopman operator, its spectral decomposition, and review PCR and RRR estimators. Section 3 describes the estimation problem and outline our main results. Section 4 presents our approach to bound eigenvalue and eigenvector estimation errors. Section 5 gives sharp upper bounds for the operator norm error. Section 6 presents our spectral learning bounds. Finally, Section 7 illustrates the implications of the bounds in practice, and is designed to provide practitioners with the tools to benchmark the performance of algorithms in real scenarios.

## 2   Background

**Dynamical Systems and Koopman Operator.** In this work we study Markovian dynamical systems, that is collections of random variables $\{X_t : t \in \mathbb{N}\}$, where $X_t$ represents the *state* at time $t$, taking values in some space $\mathcal{X}$. We focus on time-homogeneous (i.e. autonomous) systems hosting an invariant measure $\pi$ for which the *Koopman operator* [24, 27]

$$(A_\pi f)(x) := \mathbb{E}[f(X_{t+1})|X_t = x], \quad x \in \mathcal{X} \tag{1}$$

is a well defined bounded linear operator on $L^2_\pi(\mathcal{X})$, the space of square integrable functions on $\mathcal{X}$ relative to measure $\pi$. In the field of stochastic processes, (1) is also known as the *transfer operator* and returns the expected value of $f$ in the future given the present. This operator is is self-adjoint (i.e. $A_\pi = A_\pi^*$) whenever dynamics is time-reversal invariant w.r.t. $\pi$, which is satisfied by many stochastic processes in the physical sciences.

**Example 1** (Langevin Dynamics). *Let $\mathcal{X} = \mathbb{R}^d$ and let $\beta > 0$. The (overdamped) Langevin equation driven by a potential $U : \mathbb{R}^d \to \mathbb{R}$ is given by $dX_t = -\nabla U(X_t)dt + \sqrt{2\beta^{-1}}dW_t$, where $W_t$ is a Wiener process. The invariant measure of this process is the* Boltzman distribution $\pi(dx) \propto e^{-\beta U(x)}dx$, *and the associated Koopman operator is self-adjoint.*

The Langevin equation models a wealth of phenomena, such as the evolution of chemical and biological systems at thermal equilibrium [16], the mechanism regulating cell size in bacteria [1], chemical reactions [25], the dynamics of synapses [11, 42], stock market fluctuations [6] and many more. Furthermore, when $U(x) = \theta\|x\|^2/2$ $(\theta > 0)$, the Langevin equation reduces to the celebrated Ornstein–Uhlenbeck process [34, Chapter 6].

The operator (1) evolves every observable of the system forward in time. Since it is bounded and linear, it admits a *spectral decomposition*, which plays a central role in the analysis and interpretation of the dynamical system [26], as well as (nonlinear) control [2]. As in [46], to study the spectral decomposition we further assume that $A_\pi$ is a *compact* operator, which rules out the presence of

continuous and residual spectrum components and leads to

$$A_\pi = \sum_{i \in \mathbb{N}} \mu_i \, f_i \otimes f_i, \tag{2}$$

where $(\mu_i, f_i)_{i \in \mathbb{N}} \subseteq \mathbb{R} \times L_\pi^2(\mathcal{X})$ are Koopman eigenpairs, i.e. $A_\pi f_i = \mu_i f_i$. Moreover, $\lim_{i \to \infty} \mu_i = 0$ and $\{f_i\}_{i \in \mathbb{N}}$ form a complete orthonormal system of $L_\pi^2(\mathcal{X})$. In the context of molecular dynamics, the leading eigenvalues and their eigenfunctions are key in the study of long-term dynamics and so-called *meta-stable* states [see, e.g., 40].

**Koopman Operator Regression in RKHS.** Throughout the paper we let $\mathcal{H}$ be an RKHS and let $k : \mathcal{X} \times \mathcal{X} \to \mathbb{R}$ be the associated kernel function. We let $\phi : \mathcal{X} \to \mathcal{H}$ be a *feature map* [38] such that $k(x, x') = \langle \phi(x), \phi(x') \rangle$ for all $x, x' \in \mathcal{X}$. We consider RKHSs satisfying $\mathcal{H} \subset L_\pi^2(\mathcal{X})$ [38, Chapter 4.3], so that PCR and RRR approximate $A_\pi : L_\pi^2(\mathcal{X}) \to L_\pi^2(\mathcal{X})$ with an operator $G : \mathcal{H} \to \mathcal{H}$. Notice that despite $\mathcal{H} \subset L_\pi^2(\mathcal{X})$, the two spaces have different metric structures, that is for all $f, g \in \mathcal{H} \subset L_\pi^2(\mathcal{X})$, one in general has $\langle f, g \rangle_{\mathcal{H}} \neq \langle f, g \rangle_{L_\pi^2(\mathcal{X})}$. In order to handle this ambiguity, we introduce the *injection operator* $S : \mathcal{H} \to L_\pi^2(\mathcal{X})$ such that for all $f \in \mathcal{H}$, the object $Sf$ is the element of $L_\pi^2(\mathcal{X})$ which is pointwise equal to $f \in \mathcal{H}$, but endowed with the appropriate $L_\pi^2(\mathcal{X})$ norm. With this in mind, the Koopman operator restricted to $\mathcal{H}$ is simply $A_\pi S$, which is then estimated by $SG$ for some $G \in \mathrm{HS}(\mathcal{H})$. We will measure the operator norm error, $\|A_\pi S - SG\|$. This is in contrast to the more frequently used Hilbert-Schmidt (HS) norm.

Koopman operator regression estimators are supervised learning algorithms to learn the Koopman operator, in which input and output data are consecutive states of the system $(X_t, X_{t+1})$ for some $t \in \mathbb{N}$. Since the Markov process is time-homogeneous and stationary, the joint probability distribution of $(X_t, X_{t+1})$ is the same for every $t \in \mathbb{N}$ and we denote it by $\rho$. Furthermore, stationarity also implies that $X_t \sim \pi$ for all $t \in \mathbb{N}$. Given a dataset[1] $\mathcal{D}_n := (x_i, y_i)_{i=1}^n$ of consecutive states, PCR and RRR are two different strategies to minimize, under a fixed-rank constraint, the mean square error

$$\widehat{\mathcal{R}}(G) := \tfrac{1}{n} \sum_{i \in [n]} \|\phi(y_i) - G^* \phi(x_i)\|^2, \tag{3}$$

where $G \in \mathrm{HS}(\mathcal{H})$, the space of Hilbert-Schmidt operator acting on $\mathcal{H}$. PCR and RRR estimators are expressed as functions of the *input* and *cross* empirical covariances, defined respectively as

$$\widehat{C} = \tfrac{1}{n} \sum_{i \in [n]} \phi(x_i) \otimes \phi(x_i), \text{ and } \widehat{T} = \tfrac{1}{n} \sum_{i \in [n]} \phi(x_i) \otimes \phi(y_i).$$

Likewise, the population risk is $\mathcal{R}(G) = \mathbb{E}_{(X,Y) \sim \rho} \|\phi(Y) - G^* \phi(X)\|^2$, and the population covariance and cross convariance are $C = \mathbb{E}_{X \sim \pi} \phi(X) \otimes \phi(X)$, and $T = \mathbb{E}_{(X,Y) \sim \rho} \phi(X) \otimes \phi(Y)$, respectively. We note that by the *reproducing kernel property* one finds that $C = S^* S$; see e.g. [38].

**Two Important Estimators.** We next briefly recall two operator regression estimators that we study in this paper. The *Principal Component Regression* (PCR) estimator works by first projecting the input data into the $r$-dimensional principal subspace of the covariance matrix $\widehat{C}$, and then ordinary least squares are solved for such projected data, yielding the estimator [see e.g. 24]

$$\widehat{G}_{r,\gamma}^{\mathrm{PCR}} = [\![\widehat{C}_\gamma^{-1}]\!]_r \widehat{T}. \tag{4}$$

Here $\widehat{C}_\gamma := \widehat{C} + \gamma I_{\mathcal{H}}$ and $[\![\cdot]\!]_r$ denotes the $r$-truncated SVD. The population counterpart is $G_{r,\gamma}^{\mathrm{PCR}} = [\![C_\gamma^{-1}]\!]_r T$, where $C_\gamma := C + \gamma I_{\mathcal{H}}$. Note, however, that the empirical PCR estimator does *not* minimize the empirical risk (3) under the low-rank constraint.

The *Reduced Rank Regression (RRR)* algorithm, in contrast, is the *exact* minimizer of (3) under fixed rank constraint. Specifically, RRR is defined as $\widehat{G}_{r,\gamma}^{\mathrm{RRR}} := \arg\min\{\hat{\mathcal{R}}(G) + \gamma \|G\|_{\mathrm{HS}}^2 : G \in \mathrm{B}_r(\mathcal{H})\}$, where the *regularization* term $\gamma \|G\|_{\mathrm{HS}}^2$ is added to ensure stability, and $\mathrm{B}_r(\mathcal{H})$ denotes the set of bounded operators on $\mathcal{H}$ that have rank at most $r$. The closed form solution of the empirical RRR estimator is [24]

$$\widehat{G}_{r,\gamma}^{\mathrm{RRR}} = \widehat{C}_\gamma^{-1/2} [\![\widehat{C}_\gamma^{-1/2} \widehat{T}]\!]_r, \tag{5}$$

while the population counterpart is given by $G_{r,\gamma}^{\mathrm{RRR}} = C_\gamma^{-1/2} [\![C_\gamma^{-1/2} T]\!]_r$.

Once either the PCR or RRR estimators are fitted, their spectral decomposition is a proxy for the spectral decomposition of the Koopman operator $A_\pi$. Theorem 2 in [24] shows how such a decomposition can be calculated via the kernel trick for both $\widehat{G}_{r,\gamma}^{\mathrm{PCR}}$ and $\widehat{G}_{r,\gamma}^{\mathrm{RRR}}$.

---

[1] For simplicity we consider the i.i.d. setting, however our forthcoming analysis is directly applicable to sample trajectories following [24].

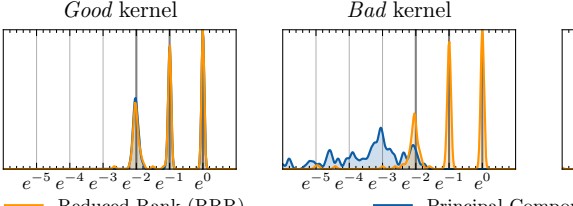

Figure 1: PCR vs. RRR in estimating the largest eigenvalues of the 1D Ornstein–Uhlenbeck process with three different kernels over 50 independent trials. Vertical lines correspond to Koopman eigenvalues. The *good* kernel is such that its $\mathcal{H}$ corresponds to the leading eigenspace of the Koopman operator, while the other two are spans of scaled and permuted eigenfunctions for which the distortion with respect to the original metric structure of $A_\pi$ introduce slow (*bad* kernel) and fast (*ugly* kernel) spectral decay of the covariance.

## 3 The Problem and Main Result in a Nutshell

In this section we introduce the spectral estimation problem, outline our main results in a distilled form, and discuss some important implications. Recall the definition of Koopman operator (1) and its spectral decomposition (2). Given a rank $r$ estimator $\widehat{G} \in B_r(\mathcal{H})$ of $A_\pi$, we let $(\widehat{\lambda}_i, \widehat{\psi}_i)_{i=1}^r$ be its spectral decomposition, satisfying $\widehat{G}\widehat{\psi}_i = \widehat{\lambda}_i\widehat{\psi}_i$. We aim to study how well a nonzero eigenvalue $\widehat{\lambda}_i$ of $\widehat{G}$ estimates its *closest* Koopman eigenvalue $\mu_{j(i)}$, where

$$j(i) = \operatorname{argmin}_{j \in \mathbb{N}} |\widehat{\lambda}_i - \mu_j|. \tag{6}$$

Moreover we wish to compare $\widehat{\psi}_i$ with the corresponding true Koopman eigenfunction. To this end, we embed $\widehat{\psi}_i$ in $L^2_\pi(\mathcal{X})$ by means of the operator $S$ and define the normalized estimated eigenfunction

$$\widehat{f}_i = S\widehat{\psi}_i \,/\, \|S\widehat{\psi}_i\|. \tag{7}$$

One of the key quantities studied in this work is the eigenvalue estimation error $|\widehat{\lambda}_i - \mu_{j(i)}|$, $i \in [r]$. Recalling that $A_\pi$ is compact and self-adjoint, the classical Davis-Kahan result [17] implies that the eigenvalue estimation error $|\widehat{\lambda}_i - \mu_{j(i)}|$ also bounds the quality of the eigenfunction approximation as

$$\|\widehat{f}_i - f_{j(i)}\|^2 \leq \frac{2|\widehat{\lambda}_i - \mu_{j(i)}|}{[\operatorname{gap}_{j(i)}(A_\pi) - |\widehat{\lambda}_i - \mu_{j(i)}|]_+} \tag{8}$$

where $\operatorname{gap}_j(A_\pi) = \min_{\ell \neq j} |\mu_\ell - \mu_j|$ is the distance between $\mu_j$ and its closest Koopman eigenvalue.

Let $\sigma_j(\cdot)$ denotes the $j$-th singular value of an operator. To give a flavour of our results, here we report spectral bounds for the Gaussian kernel. In this case, Theorem 3 below gives a high probability bound on the estimation error $|\widehat{\lambda}_i - \mu_{j(i)}|$, that is of order

$$\mathcal{O}\left(\frac{\sigma_{r+1}(A_\pi S)}{\sigma_r(A_\pi S)} + \frac{1}{\sqrt{n}}\right) \text{ for } \widehat{G}^{\text{RRR}}_{r,\gamma}, \text{ and } \mathcal{O}\left(\frac{\sigma_{r+1}(S)}{[\sigma_r(A_\pi S) - \sigma_{r+1}(S)]_+} + \frac{1}{\sqrt{n}}\right) \text{ for } \widehat{G}^{\text{PCR}}_{r,\gamma}.$$

If the Koopman operator has finite rank then $\sigma_{r+1}(A_\pi S) = 0$, the RRR estimator is unbiased, and its error goes to zero at the rate $1/\sqrt{n}$. Otherwise, recalling that $\sigma_{r+1}(S)$ is the square root of the $(r+1)$-th eigenvalue of the kernel operator [38, Chapter 4.5], if $\mathcal{H}$ is infinite dimensional $\sigma_{r+1}(S) > 0$, i.e. PCR has a strictly positive bias. In general, the presence of a bias in the estimated eigenvalues may result in the appearance of *spurious eigenvalues*. This phenomenon for PCR is well documented in practice, see e.g. [13, 14, 26, 28]. In Figure 1 we illustrate such an effect on a simple dynamical system discussed both in Example 3 and in Section 7.

## 4 Approach

The core of our analysis is Theorem 1. It reveals that in order to derive spectral estimation bounds for the Koopman operator, it is not enough to study the excess risk in the HS norm. Indeed, our spectral

bounds are determined by both the *operator norm error* of the Koopman estimator

$$\mathcal{E}(\widehat{G}) := \|A_\pi S - S\widehat{G}\|, \ \widehat{G} \in \mathrm{HS}\,(\mathcal{H}) \tag{9}$$

and the *metric distortion* between $\mathcal{H}$ and $L^2_\pi(\mathcal{X})$,

$$\eta(h) := \|h\| \, / \, \|Sh\|, \ \ h \in \mathcal{H}. \tag{10}$$

Note that since $Sh \in L^2_\pi(\mathcal{X})$ is just an equivalence class of a function $h$, $\|Sh\|$ is simply $L^2_\pi(\mathcal{X})$-norm of $h$, and hence the metric distortion can be written, with a slight abuse of notation, as $\eta(h) := \|h\|_{\mathcal{H}}/\|h\|_{L^2_\pi(\mathcal{X})}$. While the (HS norm) error was studied before [see 29, and references therein], little is know about operator norm error bounds. Moreover, the metric distortion is, to the best of our knowledge, a novel quantity in the spectral analysis of Koopman operator.

**Theorem 1.** *Let $A_\pi$ be a self-adjoint compact operator and let $r \in \mathbb{N}$. Then, for every empirical estimator $\widehat{G} \in \mathrm{B}_r(\mathcal{H})$ and every $i \in [r]$*

$$|\widehat{\lambda}_i - \mu_{j(i)}| \le \eta(\widehat{\psi}_i)\,\mathcal{E}(\widehat{G}), \quad \text{and} \quad \|\widehat{f}_i - f_{j(i)}\|^2 \le \frac{2\eta(\widehat{\psi}_i)\,\mathcal{E}(\widehat{G})}{[\mathrm{gap}_{j(i)}(A_\pi) - \eta(\widehat{\psi}_i)\,\mathcal{E}(\widehat{G})]_+}. \tag{11}$$

*Proof Sketch.* First, note that for compact self-adjoint operators $|\widehat{\lambda}_i - \mu_{j(i)}| \le \|(A_\pi - \widehat{\lambda}_i I)^{-1}\|^{-1}$. So, following the reasoning of [24, Theorem 1] and observing that $\|(A_\pi S - S\widehat{G})\widehat{\psi}_i\|/\|S\widehat{\psi}_i\| \le \mathcal{E}(\widehat{G})\eta(\widehat{\psi}_i)$ gives the right hand side of the first equation in (11). Next, since additionally $\|(A_\pi S - S\widehat{G})\widehat{\psi}_i\|/\|S\widehat{\psi}_i\| \le \mathcal{E}(\widehat{G})\eta(\widehat{\psi}_i)$, we can apply the Davis-Kahan spectral perturbation result for compact self-adjoint operators (Proposition 2, Appendix C) to bound $\sin(\widehat{\theta})$, where $\widehat{\theta}_i := \sphericalangle(\widehat{f}_i, f_{j(i)})$. The claim then follows since $\|\widehat{f}_i - f_{j(i)}\|^2 \le 2(1 - \cos(\widehat{\theta}_i)) \le 2\sin(\widehat{\theta}_i)$. The full proof can be found in Appendix C. $\qquad\square$

Note that the error (9), at least for universal kernels, can be made arbitrary small, see [24, Proposition 1]. Still, the metric distortion may dominate the error and, since the bound (11) is tight, one may have that the operator is well estimated in norm, but the estimated eigenpairs are far from the true ones. This phenomenon is at the origin of *spurious eigenvalues*. The proposed way to detect them for *deterministic* systems in [14] is to check if eigenvalue equations are satisfied empirically, which, however, is not useful for *stochastic* systems, see Rem. 4 of Appendix C.

Spuriousness may also originate from poor conditioning of the true eigenvalues, i.e. when the angle between true left and right eigenfunctions is small. Here, however, we assume $A_\pi = A_\pi^*$, so that we restrict ourselves to the case in which the *only source of spuriousness is due to the learning method*.

While we defer the discussion of the operator norm error to the next section, the following result bound the metric distortion; the proof can be found in Appendix C.

**Proposition 1.** *Let $\widehat{G} \in \mathrm{B}_r(\mathcal{H})$. For all $i \in [r]$ the metric distortion of $\widehat{\psi}_i$ can be tightly bounded as*

$$1 \, / \sqrt{\|C\|} \ \le \ \eta(\widehat{\psi}_i) \ \le \ \min(|\widehat{\lambda}_i|\,\mathrm{cond}(\widehat{\lambda}_i), \|\widehat{G}\|) \, / \, \sigma^+_{\min}(S\widehat{G}), \tag{12}$$

*where* $\mathrm{cond}(\widehat{\lambda}_i) := \|\widehat{\xi}_i\|\|\widehat{\psi}_i\|/|\langle\widehat{\psi}_i, \widehat{\xi}_i\rangle|$ *is the condition number of* $\widehat{\lambda}_i$, *and* $\widehat{\xi}_i$ *is its left eigenfunction.*

The upper bound (12) depends on the estimator's eigenvalues and their conditioning. Notice that while the true eigenvalues of $A_\pi$ have condition number one, the conditioning of the estimated ones depends on the choice of the kernel. Moreover, the upper bound can be controlled by tuning the estimator rank. Since the bound is tight (see Rem. 3 in Appendix C), the metric distortion can grow with the rank of the estimator, further motivating the use of low-rank estimators of $A_\pi$ in practice, see [26].

We end this section by introducing an empirical estimator of the metric distortion $\eta(\widehat{\psi}_i)$, given by

$$\widehat{\eta}_i := \|\widehat{\psi}_i\| \, / \, \sqrt{\langle\widehat{C}\widehat{\psi}_i, \widehat{\psi}_i\rangle}. \tag{13}$$

Proposition 4 of Appendix C shows that $\widehat{\eta}_i$ can be efficiently computed and report upper bounds for concentration around its mean. The empirical metric distortion (13), used in conjunction with the spectral bounds in Theorem 4 below, provides a proxy to assess the reliability of the PCR and RRR estimators and can be successfully used as novel model selection criterion. We refer the reader to the second and third experiment in Section 7 for concrete use-cases.

# 5  Controlling the Operator Norm Error

The HS norm error of the PCR estimator was already studied, either in the "well-specified" setting, i.e. when there exists $G_{\mathcal{H}} \in \mathrm{HS}(\mathcal{H})$ such that $A_\pi S = S G_{\mathcal{H}}$, i.e. $G_{\mathcal{H}}$ is $\pi$-a.e. Koopman operator [12, Theorem B.10]. On the other hand, KRR estimator is studied also in the "misspecified setting" [29]. But, up to our knowledge, the operator norm error has not yet been studied. To analyse these learning rates, we make the following assumptions:

**(RC)** *Regularity of $A_\pi$.* For some $\alpha \in (0,2]$ there exists $a > 0$ such that $T T^* \preceq a^2 C^{1+\alpha}$;

**(BK)** *Boundedness.* There exists $c_{\mathcal{H}} > 0$ such that $\operatorname{ess\,sup}_{x \sim \pi} \|\phi(x)\|^2 \le c_{\mathcal{H}}$, i.e. $\phi \in L_\pi^\infty(\mathcal{X}, \mathcal{H})$;

**(SD)** *Spectral Decay.* There exists $\beta \in (0,1]$ and $b > 0$ such that $\lambda_j(C) \le b\, j^{-1/\beta}$, for all $j \in J$.

While we keep assumptions **(BK)** and **(SD)** as in [18, 29], assumption **(RC)** is, up to our knowledge, novel. The rationale behind it is that for $\alpha = 1$ **(RC)** is equivalent to $\mathrm{Im}(A_\pi S) \subseteq \mathrm{Im}(S)$, in which case there exists a bounded $\pi$-a.e. Koopman operator $G_{\mathcal{H}} : \mathcal{H} \to \mathcal{H}$ [24]. On the other hand, as $\alpha \to 0$ **(RC)** becomes closer to $\mathrm{Im}(A_\pi S) \subseteq \mathrm{cl}(\mathrm{Im}(S))$ which is always satisfied for universal kernels since $\mathrm{cl}(\mathrm{Im}(S)) = L_\pi^2(\mathcal{X})$ [38, Chapter 4]. Importantly, as the next example shows, **(RC)** is weaker condition than the usual regularity conditions; see Appendix D.1 for a detailed discussion.

**Example 2.** *Let $X$ be an $\mathcal{X}$-valued random variable with law $\pi$. Consider the Markov chain $(X_t)_{t \in \mathbb{N}}$ such that $X_t = X$ for all $t \in \mathbb{N}$. Then $\pi$ is an invariant measure and $A_\pi = I_{L_\pi^2(\mathcal{X})}$ is the identity map on $L_\pi^2(\mathcal{X})$. Clearly, **(RC)** holds for all $\alpha \in (0,1]$. On the other hand, since $A_\pi S = S G_{\mathcal{H}}$ for bounded operator $G_{\mathcal{H}} = I_{\mathcal{H}} \notin \mathrm{HS}(\mathcal{H})$, HS-norm learning rates derived in [29] do not apply.*

In order to study the error of any empirical finite rank estimator $\widehat{G}$ we rely on the error decomposition

$$\mathcal{E}(\widehat{G}) \le \underbrace{\|A_\pi S - S G_\gamma\|}_{\text{regularization bias}} + \underbrace{\|S(G_\gamma - G)\|}_{\text{rank reduction bias}} + \underbrace{\|S(G - \widehat{G})\|}_{\text{estimator's variance}}, \tag{14}$$

where $G_\gamma := C_\gamma^{-1} T$ is the minimizer of the full (i.e. without rank constraint), Tikhonov regularized, HS norm error, and $G$ is the population version of the empirical estimator $\widehat{G}$.

While the last two terms in the r.h.s. of (14) depend of the estimator of choice, the first term depends only on the choice of $\mathcal{H}$ and the regularity of $A_\pi$ w.r.t. $\mathcal{H}$. In this work we focus on the classical kernel-based learning of the Koopman operator [20, 24, 29], where one chooses a universal kernel [38, Chapter 4] for which $\mathrm{Im}(A_\pi S) \subseteq \mathrm{cl}(\mathrm{Im}(S))$, and controls the regularization bias with a regularity condition. For details see Rem. 7 of Appendix D.2.

The second source of bias and the estimator's variance in our error decomposition depends on the choice of the low rank estimator. While throughout this section we consider **(RC)** for $\alpha \in [1,2]$, we discuss extensions of our results to $\alpha < 1$ in Appendix D.5.

**Theorem 2.** *Assume the operator $A_\pi$ satisfies $\sigma_r(A_\pi S) > \sigma_{r+1}(A_\pi S) \ge 0$ for some $r \in \mathbb{N}$. Let* **(SD)** *and* **(RC)** *hold for some $\beta \in (0,1]$ and $\alpha \in [1,2]$, respectively, and let $\mathrm{cl}(\mathrm{Im}(S)) = L_\pi^2(\mathcal{X})$. Let*

$$\gamma \asymp n^{-\frac{1}{\alpha+\beta}} \quad \text{and} \quad \varepsilon_n^\star := n^{-\frac{\alpha}{2(\alpha+\beta)}}. \tag{15}$$

*Let $\delta \in (0,1)$. Then, there exists a constant $c > 0$, depending only on $\mathcal{H}$, such that for large enough $n \ge r$, with probability at least $1 - \delta$ in the i.i.d. draw of $\mathcal{D}_n$ from $\rho$*

$$\mathcal{E}(\widehat{G}) \le \begin{cases} \sigma_{r+1}(A_\pi S) + c\, \varepsilon_n^\star \ln \delta^{-1} & \text{if } \widehat{G} = \widehat{G}_{r,\gamma}^{\mathrm{RRR}}, & (16a) \\[2mm] \sigma_{r+1}(S) + c\, \varepsilon_n^\star \ln \delta^{-1} & \text{if } \widehat{G} = \widehat{G}_{r,\gamma}^{\mathrm{PCR}} \text{ and } \sigma_r(S) > \sigma_{r+1}(S). & (16b) \end{cases}$$

*Proof Sketch.* The regularization bias is bounded by $a\,\gamma^{\frac{\alpha}{2}}$ by Proposition 5 of Appendix D.2. For the RRR estimator, the *rank reduction bias* is upper bounded by $\sigma_{r+1}(A_\pi S)$, while for PCR by $\sigma_{r+1}(S)$. The bounds on the variance terms critically rely on the well-known perturbation result for spectral projectors reported in Proposition 3, Appendix A. This result is then chained to two versions of the Bernstein inequality in separable Hilbert spaces. The first one is Pinelis-Sakhanenko's inequality and the second is Minsker's inequality extended to self-adjoint HS-operators, Props. 9 and 11 in Appendix D.3.1, respectively. These inequalities provide high probability bounds for the norms

of $C_\gamma^{-1/2}(\widehat{C} - C)$ and $C_\gamma^{-1/2}(\widehat{C} - C)C_\gamma^{-1/2}$, as well as $C_\gamma^{-1/2}(\widehat{T} - T)$ and $C_\gamma^{-1/2}(\widehat{T} - T)C_\gamma^{-1/2}$. Combining the bias due to regularization and variance terms, for both estimators we obtain the balancing equation $\gamma^{\frac{\alpha}{2}} = \gamma^{-\frac{\beta}{2}} n^{-\frac{1}{2}} \ln \delta^{-1}$, which yields the optimal choice of $\gamma$ and the rates. $\qquad \square$

We stress that the number of samples in the previous theorem depends on the problem's complexity, expressed in the constants $c_{\mathrm{RRR}} = \frac{1}{\sigma_r^2(A_\pi S) - \sigma_{r+1}^2(A_\pi S)}$, and $c_{\mathrm{PCR}} = \frac{1}{\sigma_r(S) - \sigma_{r+1}(S)}$. Namely, the better the separation of singular values, the smaller number of needed samples. Furthermore, analyzing the bounds (16a) and (16b), we see that faster spectral decay is, in general, preferable. For example, for the Gaussian kernel $\beta$ can be chosen arbitrarily small, yielding the rate $n^{-1/2}$. On the other hand, kernels with slow spectral decay for which $\beta = 1$ can give slower rates between $n^{-1/4}$ and $n^{-1/3}$. Finally, from the variance bounds for RRR and PCR, c.f. Appendix D.3, one can specify constants. Namely, in the slower regime when $\varepsilon_n^\star > n^{-1/2}$ we have that $c = a + 7.2 \log(10) \sqrt{2c_{\mathcal{H}}} (1 + a c_{\mathcal{H}}^{(\alpha-1)/2}) (\sqrt{c_{\mathcal{H}}} \wedge \frac{b^{\beta/2}}{\sqrt{1-\beta}})$, while in fastest regime $\varepsilon_n^\star = n^{-1/2}$, there is a significant difference between RRR and PCR since $c$ should be multiplied with the constants $c_{\mathrm{RRR}}$ and $c_{\mathrm{PCR}}$, respectively.

As argued in Section 3, the bound (16a) indicates that for rank $r$ Koopman operators the error converges to zero w.r.t. the number of training samples, while the bias of PCR is strictly positive. Hence, in order to ensure small error for PCR, high values of the rank parameter might be necessary. To theoretically explain this effect, in Theorem 6 of Appendix D.4 we give also lower bounds of operator norm error for the RRR and PCR estimators showing that $\mathcal{E}(\widehat{G}_{r,\gamma}^{\mathrm{RRR}})$ always concentrates around $\sigma_{r+1}(A_\pi S)$, while the concentration of $\mathcal{E}(\widehat{G}_{r,\gamma}^{\mathrm{PCR}})$ around $\sigma_{r+1}(S)$ depends on the *irreducible risk* of the learning problem. To illustrate the tightness of the error concentration bounds we present Example 3 (see also Appendix D.4).

**Example 3.** *Let $\mathcal{X} = \mathbb{R}$. Consider the 1D equidistant sampling of the Ornstein–Uhlenbeck process, obtained by integrating the Langevin equation of Example 1 with $\beta = 1$ and $U(x) = x^2/2$, given by $X_t = e^{-1} X_{t-1} + \sqrt{1 - e^{-2}} \epsilon_t$,, where $\{\epsilon_t\}_{t \geq 1}$ are i.i.d. standard Gaussians. For this process it is well-known [34] that $\pi$ is $\mathcal{N}(0,1)$ and that $A_\pi$ admits a spectral decomposition $(\mu_i, f_i)_{i \in \mathbb{N}}$ in terms of Hermite polynomials. We study the family of kernel functions $k_{\Pi,\nu}(x,x') := \sum_{i \in \mathbb{N}} \mu_{\Pi(i)}^{2\nu} f_i(x) f_i(x')$, where $\Pi$ is a permutation of the indices of the eigenvalues and $\nu$ is a scaling factor. The rationale behind this class of kernels is that by varying $\Pi$ and $\nu$ one morphs the original metric structure of $A_\pi$ in a way which is harder and harder to revert when learning from finite sets of data. In particular, for any target rank $r$, setting $\nu := 1/r^2$ and $\Pi$ to the permutation such that $i \mapsto 2r - i + 1$ $(i \leq r)$, $i \mapsto i - r$ $(r + 1 \leq i \leq 2r)$ and $i \mapsto i$ elsewhere, elementary algebra and our concentration bounds give*

$$|\mathcal{E}(\widehat{G}_{r,\gamma}^{\mathrm{PCR}}) - e^{-1/r}| \lesssim n^{-1/2} \ln \delta^{-1}, \qquad |\mathcal{E}(\widehat{G}_{r,\gamma}^{\mathrm{RRR}}) - e^{-r}| \lesssim n^{-1/2} \ln \delta^{-1}.$$

*We refer the reader to Figure 1 and to Section 7 for a numerical implementation of this example.*

We conclude this section with remarks on the tightness of our statistical analysis of operator norm error. Since discussed results are not the main focus of the paper, we present them in Appendix D.5.

**Remark 1** (Lower bound). *The rate $\varepsilon_n^\star = n^{-\frac{\alpha}{2(\alpha+\beta)}}$ guaranteed by (16a) matches the minimax lower bound for the operator norm error when learning finite rank $A_\pi$. Formal statement and its proof is given in Theorem 7 of Appendix D.5.*

**Remark 2** (Extension to misspecified setting). *The optimal rates for HS-norm error of the KRR estimator are developed in [29] under a stronger condition than (RC). In Theorem 9 of Appendix D.6 we extended this analysis to PCR and RRR estimators, deriving the optimal operator norm rates that also cover cases when Koopman operator cannot be properly defined as bounded operator on the chosen RKHS space $\mathcal{H}$.*

## 6  Spectral Learning Rates

Collecting all the previous results, we are now ready to present our spectral learning rates for the two estimators in a general form. For brevity, we focus on two different type of bounds in which (i) we analyse the uniform bound for the whole estimated spectra, and (ii) we express the estimators' bias in *empirical form* to provide an insight into spuriousness of eigenvalues. Moreover, we present only

eigenvalue estimation bounds, noting that the eigenfunction estimation bounds readily follow from (8). The complete results are presented in detail in Appendix E.

**Theorem 3.** *Let $A_\pi$ be a compact self-adjoint operator. Under the assumptions of Theorem 2, there exists a constant $c > 0$, depending only on $\mathcal{H}$, such that for every $\delta \in (0, 1)$, for every large enough $n \geq r$ and every $i \in [r]$ with probability at least $1 - \delta$ in the i.i.d. draw of $\mathcal{D}_n$ from $\rho$*

$$|\widehat{\lambda}_i - \mu_{j(i)}| \leq \begin{cases} \frac{2\sigma_{r+1}(A_\pi S)}{\sigma_r(A_\pi S)} + c\,\varepsilon_n^\star \ln \delta^{-1} & \text{if } \widehat{G} = \widehat{G}_{r,\gamma}^{\mathrm{RRR}}, \\ \frac{2\sigma_{r+1}(S)}{[\sigma_r(A_\pi S) - \sigma_{r+1}^\alpha(S)]_+} + c\,\varepsilon_n^\star \ln \delta^{-1} & \text{if } \widehat{G} = \widehat{G}_{r,\gamma}^{\mathrm{PCR}}. \end{cases} \tag{17}$$

The uniform eigenvalue learning rates for RRR and PCR estimators, differ in the estimator's bias. While the PCR bias has a factor $\sigma_{r+1}(S)$ in the numerator, RRR has $\sigma_{r+1}(A_\pi S) \leq \sigma_{r+1}(S)$. The striking difference happens when the Koopman operator is of finite rank. Then, assuming that $r$ is properly chosen, RRR estimator has no bias, and $c_{\mathrm{RRR}}$ is typically moderate. On the other hand, PCR's bias can be potentially large, depending of the choice of the kernel, and choosing higher rank increases $c_{\mathrm{PCR}}$, thus requiring larger sample sizes. Therefore, even in well-conditioned problems (self-adjoint operator) the *spurious eigenvalues may arise purely from the learning method*. To facilitate detection of such occurrences, we further provide an empirical estimator of the bias of both methods and illustrate their use experimentally in Section 7.

**Theorem 4.** *Under the assumptions of Theorems 2 and 3, there exists a constant $c > 0$, depending only on $\mathcal{H}$, such that for large enough $n \geq r$ and every $i \in [r]$ with probability at least $1 - \delta$ in the i.i.d. draw of $\mathcal{D}_n$ from $\rho$*

$$|\widehat{\lambda}_i - \mu_{j(i)}| \leq \begin{cases} \widehat{\eta}_i\,\sigma_{r+1}(\widehat{C}^{-1/2}\widehat{T}) + c\,\varepsilon_n^\star \ln \delta^{-1}, & \widehat{G} = \widehat{G}_{r,\gamma}^{\mathrm{RRR}}, \\ \widehat{\eta}_i\,\sqrt{\sigma_{r+1}(\widehat{C})} + c\,\varepsilon_n^\star \ln \delta^{-1}, & \widehat{G} = \widehat{G}_{r,\gamma}^{\mathrm{PCR}}. \end{cases} \tag{18}$$

We remark that when $A_\pi$ is of finite rank $r$, the bound above for the RRR estimator reduces to

$$|\widehat{\lambda}_i - \mu_i| \leq c\,\varepsilon_n^\star \ln \delta^{-1} \ \text{ and } \ \|\widehat{f}_i - f_i\|^2 \leq \frac{2\,c\,\varepsilon_n^\star \ln \delta^{-1}}{[\mathrm{gap}_i(\widehat{G}_{r,\gamma}^{\mathrm{RRR}}) - 3\,c\,\varepsilon_n^\star \ln \delta^{-1}]_+},$$

see Cor. 1 in Appendix E. Hence, in this case RRR algorithm can learn all the eigenvalues and eigenfunctions of $A_\pi$ with rate $\varepsilon_n^\star = n^{-\frac{\alpha}{2(\alpha+\beta)}}$. On the other hand, even in this case, the bounds for the PCR estimator do not guarantee unbiased estimation of Koopman eigenvalues and eigenfunctions.

**Choosing $\gamma$ and $r$.** The bias term $\sigma_{r+1}(A_\pi S)/\sigma_r(A_\pi S)$ appearing in (17) represents the theoretical limit when estimating eigenvalues using RRR. It reflects the capacity of the RKHS to detect the separation of the leading $r$ Koopman eigenvalues from the rest of its spectra. If $A_\pi$ has infinite rank and slowly decaying eigenvalues, estimating the leading ones becomes challenging, since increasing $r$ leads to smaller operator norm error, but larger bias. Luckily, in many practical problems there is a separation of time-scales in the dynamics and the above ratio can be controlled by choosing $r$ appropriately. While we do not have access to $A_\pi S$, we can still choose $r$ via the empirical operator $\widehat{C}^{-1/2}\widehat{T}$, see Proposition 20 of Appendix D.4. Note also that the optimal $\gamma$ depends on $\alpha$ which is typically unknown. In practice, one can implement a standard grid-search CV procedure for time series to tune this parameter.

**Spectral Bias as a Tool for Model Selection.** In equation (18), the data dependent quantities $\widehat{s}_i(\widehat{G}_{r,\gamma}^{\mathrm{RRR}}) := \widehat{\eta}_i\,\sigma_{r+1}(\widehat{C}^{-1/2}\widehat{T})$ and $\widehat{s}_i(\widehat{G}_{r,\gamma}^{\mathrm{PCR}}) := \widehat{\eta}_i\,\sigma_{r+1}(\widehat{C})$ represent the *empirical spectral biases* of RRR and PCR estimators of the Koopman operator, respectively. When they are small enough, the spectral estimation error is dominated by the same variance term, which decreases as the number of samples grows. Therefore, given a number of different kernels, we propose to select the best one (w.r.t. spectral estimation) by choosing the smallest spectral bias. This is illustrated in the Alenine Dipeptide example of following section.

**Normal operators.** Since Davis-Kahan theorem [17] also holds for normal operators, the results in this section apply whenever $A_\pi A_\pi^* = A_\pi^* A_\pi$. While in this case Koopman eigenfunctions remain orthogonal in $L_\pi^2(\mathcal{X})$, the eigenvalues are in general complex. On the other hand, extension beyond normal compact operators asks for involved spectral perturbation analysis and a new statistical learning theory.

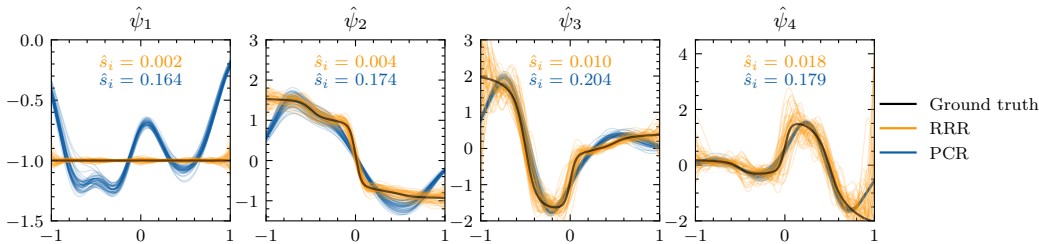

Figure 2: Estimated eigenfunctions $\widehat{\psi}_i$ of a Langevin dynamics vs. ground truth. The average empirical biases $\widehat{s}_i$, $i \in [4]$ are discussed at the end of Section 6. The results correspond to 50 independent estimations on 2000 training points each. PCR and RRR estimators were fitted with the same parameters: Gaussian kernel of length scale 0.175, $\gamma = 10^{-5}$ and $r = 4$.

## 7 Experiments

We illustrate various aspects of our theory with simple experiments. They have been implemented in Python using the library Kooplearn (available at `https://github.com/CSML-IIT-UCL/kooplearn`) to fit the PCR and RRR estimators. Full details are in Appendix F.

**Learning the Spectrum of the Ornstein–Uhlenbeck Process.** In this experiment we designed three different kernel functions (the "*good*", the "*bad*" and the "*ugly*") to illustrate how an unseemly kernel choice can induce catastrophic biases in the estimation of Koopman eigenvalues. We focus on the uniformly sampled Ornstein-Uhlenbeck (OU) process, discussed in Example 3, relying on the spectral decomposition of its Koopman operator $(\mu_i, f_i)_{i \in \mathbb{N}}$ to design the three kernel functions. The *good* kernel is just the sum of the leading $T = 53$ terms of the spectral decomposition of $A_\pi$, i.e. $k_{\text{good}}(x, y) := \sum_{i=1}^{T} \mu_i f_i(x) f_i(y)$. The associated RKHS coincides with the leading eigenspace of $A_\pi$, and *no deformation of the metric structure* takes place, so that the injection map $S \colon \mathcal{H} \hookrightarrow L^2_\pi(\mathcal{X})$ is a partial isometry. The *bad* kernel is defined according to the construction presented in Example 3 for $\nu = 1/r^2$ where $r$ is the rank of the estimator. For this kernel, the introduced bias is innocuous for RRR, but lethal for PCR. Finally, the *ugly* kernel corresponds to $\nu = r^2$, introducing large quotients $\sigma_{r+1}(A_\pi S)/\sigma_r(A_\pi S)$ and $\sigma_{r+1}(S)/\sigma_r(S)$, and, hence, an irreparable bias in both estimators.

Figure 1 depicts the distribution of the eigenvalues estimated by PCR and RRR over 50 independent simulations, against the ground truth. For both algorithms each simulation is comprised of 20000 training points, the regularization is $\gamma = 10^{-4}$ and the rank is $r = 3$. The three largest eigenvalues of $A_\pi$ are correctly estimated by both algorithms for $k_{\text{good}}$ and by RRR for $k_{\text{bad}}$. On the contrary, the distribution of the eigenvalues for $k_{\text{ugly}}$ (and $k_{\text{bad}}$ for PCR) does not concentrate around any true eigenvalue of $A_\pi$, signaling the presence of *spurious eigenvalues* in the estimation.

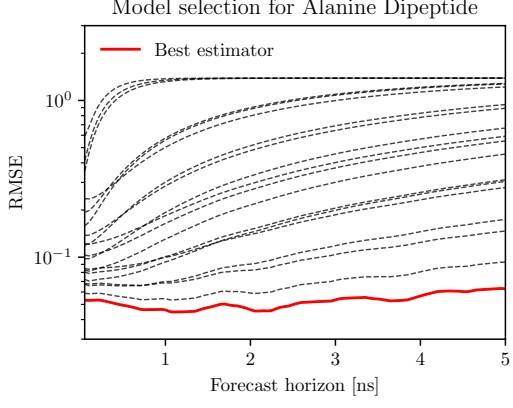

Figure 3: Forecasting RMSE on the Alanine Dipeptide dataset for 19 different RRR estimators, each corresponding to a different kernel, which show how the best model, according to the empirical spectral bias metric, also attains the best forecasting performances by a large margin.

**A Realistic Example: Langevin Dynamics.** Because of its ubiquitous use in modelling real systems, we now study a numerical implementation of the Langevin dynamics Example 1 with $\beta = 1$ and a potential $U(x) = 4(x^8 + 0.8e^{-80x^2} + 0.2e^{-80(x-0.5)^2} + 0.5e^{-40(x+0.5)^2})$ that is a mixture of three Gaussians barriers at $x \in \{-0.5, 0, 0.5\}$ and a smooth "bounding" term $\propto x^8$ constraining most of the equilibrium distribution in the interval $[-1, 1]$, see [37]. In Figure 2, for $i \in [r]$, we compare the $\widehat{f}_i$ estimated by PCR and RRR against the ground truth $f_i$. The visible difficulty of PCR compared to RRR in estimating eigenfunctions is nicely explained by larger values of the empirical bias for PCR, which we report in the upper part of the figure. The reference eigenpairs of $A_\pi$ have been obtained by diagonalizing a finely discretized approximation of the infinitesimal generator (see Appendix A).

**Spectral Bias and Model Selection: the Case of Alanine Dipeptide.** In this example we show that minimizing the first term on the r.h.s. of (18) over a validation dataset, is also a good criterion for Koopman model selection. We use a realistic simulation of the small molecule Alanine Dipeptide already discussed in [24, 44]. We trained 19 RRR estimators each corresponding to a different kernel and then we evaluated the forecasting RMSE on 2000 initial conditions drawn from a test dataset. In Figure 3 we report these errors, highlighting the model with the smallest average empirical spectral bias (18) evaluated on 5000 validation points.

# 8 Conclusion

We established minimax optimal rates for the operator norm error in the Koopman regression problem, which we then used to derive sharp estimation bounds for eigenvalues and eigenfunctions of the Koopman operator associated with a time-invariant Markov chain. We considered two important estimators that implement either principal component regression (PCR) or reduced rank regression (RRR) to learn a linear operator on a reproducing kernel Hilbert space. Our bounds indicate that RRR may be advantageous over PCR (also known as EDMD, the de-facto estimator in the data-driven dynamical system literature) which may exhibit a larger estimation bias. This ultimately depends on the choice of the kernel, which significantly impacts the rate. A bad choice of the kernel could also introduce spurious eigenvalues, a phenomena which has been observed in the literature and which is now explained by our theory. Finally, we proposed a method to detect spuriousness in practice, which can be used also as a kernel selection tool. A limitation of this work is that it applies to compact normal operators only. While many real dynamical systems involve such operators, in the future our analysis may be extended using more sophisticated spectral perturbation theory.

**Acknowledgements.** This work was supported in part from the PNRR MUR Project PE000013 CUP J53C22003010006 "Future Artificial Intelligence Research (FAIR)", funded by the European Union – NextGenerationEU, and EU Project ELIAS under grant agreement No. 101120237.

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
