# OpenReview forum: "Sharp Spectral Rates for Koopman Operator Learning"
_NeurIPS.cc/2023/Conference — NeurIPS 2023 spotlight_

### Official Review · Reviewer_ZSn2 · 2023-06-22

**Soundness:** 3 good
**Presentation:** 4 excellent
**Contribution:** 3 good
**Rating:** 7
**Confidence:** 3

**Summary:**

This paper develops bounds on how close the approximated Koopman modes and eigenvalues are to the true eigenvalues and modes, for two important classes of methods used to compute the Koopman mode decomposition. The authors find that one class, Principal Component Regression, which included extended dynamic mode decomposition (EDMD), can suffer more from poorly chosen kernels and can have larger bias than Reduced Rank Regression (RRR). They additionally provide an empirical method for determining spurious eigenvalues, which can be used for model selection.

**Strengths:**

1. This paper is well written and easy to follow.

2. This paper provides new techniques for computing bounds on the approximated Koopman spectral objects, and the discovery that PCR can have larger bias than RRR, are important ones for the field.

3. This paper provides a new empirical method for identifying spurious eigenvalues and making model selection. Again, both of these are important topics for the field and for the application of numerical methods in applied settings.

4. The numerical examples provided in Figs. 1-3 are helpful for understanding the theory developed, and provide support for the developed claims.



**Weaknesses:**

1. Klus et al., 2016 and Korda and Mezic, 2018, as examples, proved the convergence of EDMD to the true Koopman operator, when $M \rightarrow \infty$, where $M$ is the number of data points. This work seems to be missing from the paper, as does discussion surrounding how the paper differs from that work. I assume the primary difference is that this paper's results are not in the asymptotic limit (although, they are in the sense that the bounds for RRR, for example, in the Gaussian case converge as $1/\sqrt{n}$, which approaches $0$ as $n\rightarrow \infty$). Additionally, the results for PCR obtained by this paper would suggest that in the asymptotic limit EDMD does not converge, since it has a bias. Discussion on how this is reconciled with the work of Klus et al., 2016 and Korda and Mezic, 2018, is necessary.

2. Fig. 3 was confusing. Was the best estimator found on the test data set, and then the red line in Fig. 3 the result of applying it to the validation data? A secondary panel in that figure describing what was being done would be helpful.

MINOR COMMENTS:

1. It was unclear to me how $| \lambda_i - \mu_{j(i)} | \leq || (A_\pi - \lambda_i I)^{-1}||^{-1}$ leads to observing that $||(A_\pi  - S\hat{G})\hat{\psi}_i|| \leq \mathcal{E}(\hat{G})\eta(\hat{\psi})$ (lines 154-155). Adding a little more detail/comment on this would be helpful.

2. "left hand side" (line 156) should be "right hand side" no?

3. The connection between DMD and KMD should be discussed (lines 22-25) (Rowley et al., 2009).

4. The original EDMD paper (Williams et al., 2015) should be cited when discussing EDMD for the first time (line 31).

5. Very minor but both "non-linear" and "nonlinear" are written.

**Questions:**

1. How does this work compare to previous work studying the convergence of EDMD (e.g., Klus et al., 2016; Korda and Mezic, 2018)?

2. What exactly is Fig. 2 showing (in terms of details)?

**Limitations:**

The authors did a good job being clear that their work was limited to self-adjoint operators.

---

> ### Author Rebuttal · Authors · 2023-08-09
>
> We appreciate the reviewer's insightful evaluation and valuable comments. In what follows, we aim to address the highlighted weaknesses and respond to the reviewer's questions.
>
> __Weakness 1 & Question 1:__
>
> First, we would like to stress out that [Korda and Mezic 2018] prove the consistency (infinitely many samples $n\to\infty$) of EDMD with $L^2_\pi$ orthogonal projection of the Koopman operator onto the finite $r$-dimensional subspace spanned by the dictionary of functions. Note that this _projection is not the same as the Koopman operator_ (acting on the whole $L^2_\pi$), see pg. 50 of [8]. Additionally, [Korda and Mezic 2018]  show that EDMD is consistent with the Koopman operator on $L^2_\pi$, only when, under certain assumptions,  $r\to\infty$. Moreover, these results are limited to deterministic dynamical systems. On the other hand, [Klus et al. 2016] prove the consistency of EDMD for stochastic processes with Galerkin projection on the same subspace (finite $r$), while our results also imply consistency of EDMD when $r\to\infty$. Note that our work is concerned with the convergence of the spectral decomposition, and that the eigenvalues of (Galerkin) projections, depending on the choice of the subspace, may be spurious for finite $r$. In light of these considerations we do not see any contrast between our results and the existing literature on the subject.
>
> Next, we would like to recall that proving consistency of estimators (an infinite number of data points) is typically easier than proving non-asymptotic generalization bounds for a finite number of data points, especially when the goal is to establish the matching information theoretic lower bound. This provides deeper insight into the learning problem. When one deals with eigenvalues, this becomes even more so, since, in general, the eigenvalue estimation hides the effect of sensitivity to perturbations that cannot be seen from the asymptotic results, but needs more sophisticated perturbation analysis. Namely, even though the estimator is consistent (hence its eigenvalues converge to the true ones) spurious eigenvalues may arise for finite sample size. This happens when the norm of the estimator error is small but the estimator’s eigenvalues may be very far from the true ones.
>
> __Weakness 2:__
>
> Figure three has been produced as follows: we trained every one of the 19 estimators considered on the same train dataset of 2000 points. Then, for each of the fitted models we have evaluated the (data-dependent) spectral bias introduced in lines 291-295 related to the r.h.s. of Eq. (18). The evaluation of the spectral bias has been done on a validation set comprising 5000 points. From this evaluation procedure we were able to compare the 19 models, rank them, and select the “best” one as the one minimizing the spectral bias. Finally, we have selected 2000 trajectories out of a third (independent) test set, and evaluated the RMSE attained by each model in forecasting the state. We will clarify this procedure either in the text or with an additional panel as suggested.
>
> __Question 2:__
>
> For 50 trials and n=2000 training points both PCR and RRR estimators were fitted with the same parameters, in particular with rank r=4. Then, the estimated eigenfunctions using PCR (blue) and RRR (orange) were plotted, comparing them to the true eigenfunctions in black (one per plot). Additionally, average (across 50 independent trials) empirical spectral biases $\hat{s}_1,...,\hat{s}_4$ were computed. Comparing the values of $\hat{s}_i$ with the eigenfunction estimation error, we observe that, as indicated by our theory, spectral biases predicts the performance of the estimators (PCR vs RRR) on recovering Koopman eigenfunctions. We will revise the caption to help the reader better understand the Figure.
>
> __Minor comments:__
>
> 1. We are sorry for the confusion, lines 155-156 should read “So, following the reasoning of [24 , Thm. 1] and observing that $\Vert(A_\pi S- S\hat{G})\hat{\psi}\_i\Vert / \Vert S\hat{\psi}\_i \Vert \leq \mathcal{E}(\hat{G})\eta(\hat{\psi}\_i)$ gives the left hand side of (11) .” Namely, recalling that $\Vert (A_\pi-\hat{\lambda}\_i I)\^{-1}\Vert\^{-1}= \min_{\Vert f\Vert =1} \Vert A_\pi f-\hat{\lambda}_i f \Vert$ and plugging $\hat{f_i}$ instead of $f$, we obtain (11).
> 2. Sorry for the confusion, it should state: “r.h.s. of the first equation in (11)”
> 3. Indeed, we will revise the lines 22-26, citing  [Rowley et al. 2009], to clarify that DMD is an algorithm to estimate the Koopman mode decomposition (KMD).
> 4. Yes, thank you!
> 5. Thanks for spotting it, we will fix it.

---

> > ### Comment · Reviewer_ZSn2 · 2023-08-11
> > **Response to reviewers**
> >
> > Thank you for your detailed rebuttal.
> >
> > These responses, as well as the reviews from the other reviewers (and the authors' responses to those reviewers) make me confident that this is a strong paper with good contributions. The changes/clarifications the authors propose to make in the revised version of the manuscript will further increase its quality.
> >
> > I will therefore increase my score.

---

### Official Review · Reviewer_B3rh · 2023-07-07

**Soundness:** 4 excellent
**Presentation:** 4 excellent
**Contribution:** 4 excellent
**Rating:** 9
**Confidence:** 3

**Summary:**

This paper studies the approximation and learning of the Koopman operator. Koopman operator is helpful in modeling a broad class of Markovian dynamical systems. In this paper, two types of approximation strategies are studied, namely, "Principal Component Regression (PCR)" and "Reduced Rank Regression (RRR)." Both types are based on general reproducing kernel Hilbert spaces. With the assumptions that the population covariance operator bounds the population cross covariance operator, the RKHS feature map being L-infinity, and the eigenvalues of the population covariance operator decay as O(i^(-1/beta)) with 0<beta<=1, the operator norm error, the eigenvalue estimation error, and the eigenfunction approximation error are bounded.

**Strengths:**

The paper's originality is high, because of the new error estimation provided. This paper represents solid research results with high quality. The writing is clear and easy to follow. The paper's significance is guaranteed by the broad scope of applications of the Markovian dynamical systems, including Langevin dynamics.

**Weaknesses:**

We do not have significant concerns about this paper. No weakness is identified.

**Questions:**

1. Lines 116--117: What is the definition of rank-r operator? If the dimension of the image of one operator is finite, the operator is automatically Hilbert-Schmidt. If this is the definition, I guess using the notion of "rank-r Hilbert-Schmidt operator" is confusing because it implies the possibility of rank-r operators that are not Hilbert-Schmidt.
2. Which theorem does it refer to for the claim in Line 335?
3. In Line 68, the Koopman operator A_pi is assumed self-adjoint, and in Line 81, A_pi is further assumed compact. Is the second assumption made globally for the whole paper? In Theorem 3, A_pi is again assumed compact and self-adjoint, but this assumption does not appear in Theorem 4. This could not be very clear, leaving the reader wondering whether this assumption is adopted for Theorem 4.

**Limitations:**

No limitation issues were found.

---

> ### Author Rebuttal · Authors · 2023-08-09
>
> We appreciate the reviewer's insightful evaluation and the positive comments. In what follows, we aim to respond to the reviewer's questions.
>
> __Questions:__
>
> 1. We agree, the notation is not the best. We will follow the advice and denote by ${\rm B}_r(\mathcal{H})$ finite rank operators $\mathcal{H}\to\mathcal{H}$.
> 2. It is Theorem 7 in the appendix that is discussed in lines 252-254. Should we include this Theorem on the lower bound in the main body?
> 3. Thank you for spotting this. It is a typo, the statement of Theorem 4 should start as “Under the assumptions of Theorems 2 and 3…”

---

> > ### Comment · Reviewer_B3rh · 2023-08-16
> > **Including key theorem into main body**
> >
> > For 2, since it is claimed that "We established minimax optimal rates for the operator norm error in the Koopman regression problem" in line 335, I think it makes sense to include the relative theorem(s) to the main body. Of course, the proofs could be placed in the appendix.

---

### Official Review · Reviewer_nHa6 · 2023-07-18

**Soundness:** 3 good
**Presentation:** 3 good
**Contribution:** 3 good
**Rating:** 6
**Confidence:** 2

**Summary:**

The authors provide bounds for the spectral decomposition of the estimated Koopman operators. The bounds are given for self-adjoint time-reversible operators in terms of two new metrics. Compared to estimation guarantees given in terms of the Hilbert Space distance norm, the proposed bounds require less restrictive assumptions. The theoretical results are specialized for two existing estimation algorithms.


**Strengths:**

Estimating the eigenvalue of the Koopaman operator is a widespread problem. And the work seems to extend to general self-adjoint operators. The proposed new evaluation metrics are interesting, especially if they produce theoretical bounds that hold under less restrictive assumptions.

**Weaknesses:**

It is unclear how the bounds depend on the sample size and why it is interesting to specialize them for existing estimators. The experiments show the performance of two existing estimators but have no straightforward link with the theoretical part of the paper. It
would be more interesting to plot, for a given estimator, the wideness of the proposed bounds versus the (a posteriori) empirical estimation error.

After assuming that the process is time-homogeneous and stationary, the learning task looks similar to standard non-parametric regression. The authors should specify what are the challenging aspects of the dynamical setup. Otherwise, the contribution of the paper is unclear. The core part of this work seems to be applying classical spectral bounds to a finite-dimensional approximation of HS operators. If the novelty is to use new metrics, the paper should focus more on explaining why these new metrics are better than the HS norm.

The paper's conclusion is somehow expected. Direct learning of low-rank representations is better than projecting the data and solving an unconstrained problem. The latter option may have computational advantages. But the authors do not comment on it.


**Questions:**

- I do not fully understand this sentence, "The Koopman operator [...], and DMD relies [...] on to, in turn, estimate its spectral decomposition."

-The Koppman operator is a linear HS operator. What is peculiar about bounding the Koopman operator compared to other HS operators?

- How common and restrictive are the assumptions that the operator is time-reversible and compact?

- Why do you say that "the empirical PCR estimator does not minimize the empirical risk (3) under the low-rank constraint."? Is this because of the projection? Or because the optimization is unconstrained?

- How can the learning method generate spurious eigenvalues? Is the standard operator norm insensitive to eigenvalue spuriousness?

- Is the regularity condition called RC on page 5 used in the bounds? I am confused by the sentence on the following page, "where one chooses a universal kernel for which ...".

- Intuitively, bounding the error on the estimated eigenvalue can be easier than on the estimated error functions or eigenvector. I understand the final formula may be too intricate for the main text. But it would be helpful for the reader to see an intuitive explanation of the claim, "the eigenfunction estimation bounds readily follow from (8)".


**Limitations:**

The authors say that restricting the analysis to self-adjoint operators is the main limitation of their work. This is mentioned in the very last lines of the paper. It would be better to elaborate on this limitation in the introduction, where the self-adjoint assumption is made.

---

> ### Author Rebuttal · Authors · 2023-08-09
>
> We appreciate the reviewer's insightful evaluation and valuable comments. In what follows, we first aim to address the weaknesses and respond to the reviewer's questions.
>
> __Weaknesses:__
>
> >_It is unclear..._
>
> We are not sure to understand the reviewer's remark. In the appendix all the bounds are expressed explicitly, even including all relevant constants. In the main text, for easier readability, they are expressed as learning rates, that is in the standard form commonly used in statistical learning papers where the asymptotic dependence on the number of samples and the probability is shown.
>
> >_The experiments [...] no straightforward link..._
>
> We respectfully disagree. Each experiment is directly linked to theoretical results. Please note that:
> - The experiment on the Ornstein–Uhlenbeck Process is an illustration of Example 3 that answers why the bias term in Thm. 2 is tightly estimated. This is remarked throughout the paper, e.g. in lines 232-238
> - The experiment on Langevin Dynamics is used to show two aspects. First, in the main text we show that the novel object, the empirical spectral bias $s_i$ introduced in Thm. 4 and lines 291-29, predicts well the quality of the estimated eigenfunctions. Second, in the App. F, we empirically verify (on a validation set) the spectral learning rates for eigenvalues and eigenfunctions.
> - The experiment on Alanine Dipeptide demonstrates how our theoretical results on the empirical spectral bias can be used as a tool for model selection, an important practical task within kernel-based Koopman methods for dynamical systems.
>
> >_...empirical estimation error_
>
> First, we would like to stress that we have derived minimax optimal learning rates which offer a framework for objectively comparing different methods. In addition to the theoretical analysis, we have properly verified the bounds on a validation set, see Appendix F.
>
> >_After assuming [...] the learning task…_
>
> As introduced in [24], the learning task we consider involves estimating spectral decomposition of an unknown linear operator on an unknown domain, leading to novel challenges. See, please also the global response to all reviewers, where we elaborate on different challenges and aspects of our contributions.
>
> >_The paper's conclusion..._
>
> We agree, the RRR estimator proposed in [24], was shown to be superior to the standard PCR in terms of uniform learning bounds, and computational complexity was studied, too. Still, we deem our detailed analysis essential to fully comprehend the non-trivial phenomena arising when learning the spectral decomposition of Koopman operators, crucially important in dynamical systems applications, as stated in e.g. [8, 9, 26, 36].
>
> __Questions:__
>
> - The interest in the Koopman operator stems from its ability to globally linearize nonlinear dynamical systems in a (typically infinite dimensional) function space, usually called space of observables. In this new space one can solve a linear dynamical system via spectral decomposition, and the result in the original (state) space is known as Koopman mode decomposition (KMD). KMD is an efficient formula for evolving general observables (measurements) of nonlinear dynamical systems, while Dynamic Mode Decomposition (DMD) is a method to approximate KMD from the recorded trajectory using a space of linear functions as observables.
> - We are not sure to fully understand the reviewer’s question. Should they please elaborate on it? Please note that the Koopman operator is in general only bounded. In order for it to also be HS, the transition kernel needs to have square integrable density w.r.t. the invariant distribution. On the other hand, the restriction of the Koopman operator onto RKHS, i.e. $A_\pi S$, is an HS-operator. Concerning the utility of the Koopman operator please see the discussion in e.g. [7, 8, 9, 22, 23, 26, 31].
> - We tried to explain better the assumptions and contributions in the global reply to all reviewers. Compactness of $A_\pi$ is guaranteed, e.g., whenever the transition kernel is absolutely continuous w.r.t. the invariant measure, i.e. $p(x,\cdot)<< \pi$ for $\pi$-a.e. $x\in\mathcal{X}$. This is a reasonable assumption implying that the system cannot transition into sets that at equilibrium have measure zero. If additionally, this density $f(x,y):= \frac{dp(x,\cdot)} { d\pi}(y)$, $x,y\in\mathcal{X}$ is symmetric, i.e. $f(x,y)=f(y,x)$ $\pi$-a.e., then the Koopman operator $A_\pi$ is self-adjoint. Many physical systems satisfy these conditions, see lines 74-77.
> - PCR estimator $\hat{G}$ is not a minimizer of objective (3) under the rank constraint $rank(\hat{G})\leq r$, instead it is a spectral filter applied to the OLS solution.
> - In general, spuriousness of eigenvalues is a perturbation phenomena happening when $|\lambda_i(A) - \lambda_i(A+\Delta)|>>\Vert \Delta \Vert$. The norm used to properly address this phenomena is the operator norm, under which the equality holds for normal operators. Hence, in our case the spuriousness stems only from the learning method, since we study the _operator norm_ error in the estimation of normal $A_\pi$. Since the estimator is built from a subspace $\mathcal{H}$ of a function space $L^2_\pi(\mathcal{X})$ there is a metric distortion that comes into play. This doesn’t occur in the classical estimation of spectra in (finite-dimensional) matrix problems.
> - Yes, all the results in the bounds use (RC) condition. The sentence means that when the kernel is universal, the only contribution to the regularization bias is due to $\gamma$ thanks to (RC). For non-universal kernels, an additional term is present, see Prop 5 of App D.2.
> - Thanks! Indeed, in general, bounding errors for eigenvectors is more difficult than for eigenvalues. However, for normal linear operators, the situation is easier due to Davis-Kahan theorem (Prop 2 of App A.1). As you have suggested we will elaborate better on this in the revised manuscript.

---

> > ### Comment · Reviewer_nHa6 · 2023-08-11
> > **I agree with the other reviewers and raise my score to Accept.**
> >
> > Many thanks to the authors for the detailed rebuttal. Probably, I should have read the appendix more carefully. Thank you for explaining the difference between Koopman and HS operators. The authors' answers also address my concern about the connection between theory and experiments.

---

### Official Review · Reviewer_GGn4 · 2023-07-24

**Soundness:** 3 good
**Presentation:** 4 excellent
**Contribution:** 3 good
**Rating:** 7
**Confidence:** 3

**Summary:**

This paper analyses two approximation techniques for self-adjoint compact Koopman operators. Both estimators are constructed based on regularized least-squares with a certain rank condition in mind. The Principal Component regression (PCR) performs unconstrained Tikhonov   estimation of data projected on a low-dimensional subspace. The Reduced Rank Regression (RRR) minimizes the Tikhonov objective function subject to a rank constraint on the optimizer. Both estimators admit closed form solutions. The paper study spectral properties of these estimators, specifically deviation of eigenvalues and eigenvectors. The novelty of this study resides on the error measures: operator norm error of the Koopman operator estimation (instead of Hilbert-Schmidt) and the "metric distortion" that compares the two Hilbert space, the RKHS that approximates the process, and the ambient L^2 Hilbert space where the Koopman operator is initially defined.
The conclusion of this study is that both estimators have a similar variance, but the PCR may have a potentially larger bias, particularly for badly chosen kernels.


**Strengths:**

The authors employ existing state-of-the-art bounds in spectral theory of compact operators. The study sheds light on the phenomenon of "spurious eigenvalues". The asymptotic rates of convergence are shown to be optimal.
Overall it seems a solid paper.

**Weaknesses:**

The rates of convergence and error bounds are tight, but only asymptotically. Since both PCR and RRR have similar asymptotic rates (for variance), the constants become important.  A more careful analysis of the constants would strengthen the paper. However it is understandable that such a study might be analytically too complex.

The authors mention that results are limited to self-adjoint Koopman operators. This is true, however, compact operators admit a SVD factorization with similar spectral properties (control of singular values) as self-adjoint operators. At a cursory reading, the results obtained in this study seem extendable to non-self-adjoint compact Koopman operators.


**Questions:**

1. The paper is well-written. Just a few typos that can be easily fixed. I suppose in Example 3, the definition of the permutation Pi, the first case is i<= r, right?

2. I understand that condition (RC) is weaker than Im (A_pi S)\subset Im(S). Is it weaker also than Im (A_pi S)\subset closure(Im(S)), with closure w.r.t. L^2-norm ?

3. It would be useful to indicate the Hilbert space norms throughout the paper. Particularly in equation (10), but also elsewhere.

**Limitations:**

Nothing to be reported here.

---

> ### Author Rebuttal · Authors · 2023-08-09
>
> We appreciate the reviewer's insightful evaluation and valuable comments. In what follows, we aim to address the highlighted weaknesses and respond to the reviewer's questions.
>
> __Weaknesses:__
> - Regarding the variance bounds' constants, we carefully derived the proofs in the Appendix so that they can be easily analytically derived and compared using the right-hand side (r.h.s.) of Equation (56) for RRR and Equation (57) for PCR. Indeed, since both equations are combined with the same regularization bias term containing $\gamma$, they are sufficient to understand differences in constants. As suggested, in the revised version we will address the constants, too. For the reviewer’s convenience, here we report the most important findings. Interestingly, one can see that when $\varepsilon_n^1(n^{-\frac{1}{\alpha+\beta}}, \delta/5) < 0.1$, for notation see (46) in Apendix D.3.1, we have that both estimators have the same constant in front of the leading term $n^{-\frac{\alpha}{2(\alpha+\beta)}} \log(1/ \delta)$ which is equal to $$ a+7.2 \log(10)\sqrt{2 c_{\mathcal{H}}} (1+ a c_{\mathcal{H}}^{(\alpha-1)/2}) \,b^{\beta/2} / \sqrt{1-\beta},$$ where all appearing constants are defined in assumptions (BD), (SD) and (RC). However, It is worth noting that in front of the (in general) slower term of the order $n^{-1/2}$ there is an important difference in the constants expressed as $c_{\rm PCR}$ vs $c_{\rm RRR}$ (both defined in line 226). For example, for the Gaussian kernel and self-adjoint Koopman operator such that $\alpha>1$, the leading constant becomes simply $\approx 48$, while for maximal $\alpha$ being equal to 1, we have that constant for PCR becomes $const. c_{PCR}  \sqrt{\log(tr(C) / \Vert C \Vert},$ _const._ being a universal, while the one for RRR is obtained by replacing $c_{PCR}$ by $c_{RRR}$.
>
> - In the reply to all the reviewers we tried to clarify this aspect. As long as one does not analyze the spectral rates, compactness and self-adjointness are not really needed and the boundedness of the Koopman operator suffices as long as (BD), (SD) and (RC) hold. Concerning the SVD, we recall that, in general, singular value decomposition is related to the notion of norms, while eigenvalue decomposition is related to powering of operators (hence, dynamics). Consequently, the singular values of non-normal compact Koopman operators do not provide useful information about the system's evolution. That being said, we have already established the rates for estimating the singular values of $A_\pi S$ (the restriction of the Koopman operator to the RKHS) via estimator $\hat{B} :=\hat{C}_\gamma^{-1/2} \hat{T}$. This result is presented in Propositions 6 and 17 of the Appendix D and serves as a key element in deriving bounds for the RRR estimator. As the reviewer suggests, it might be interesting to further analyze the estimation of the SVD of compact Koopman operators. We leave this as a direction of future work as it looks like a nontrivial open question.
>
>
> __Questions:__
> 1. Yes, $i\leq r$, thanks for spotting it.
> 2. (RC) condition is not weaker than $Im(A_\pi S)\subseteq cl(Im(S))$ which is always assured when the kernel is universal. Let's call the latter (UC). In a certain sense, letting the $\alpha\to0$ (RC) condition “gets closer” to (UC) $\implies$ (RC). To be precise, we refer to Appendix D.1 that contains details on the relationships between the assumptions. There we see that using Sobolev norms we can define interpolation spaces $[\mathcal{H}]\_{\alpha}$ that progressively cover $L^2_\pi(\mathcal{X})= [\mathcal{H}]\_0$, as $\alpha\to 0$. Thus, since  $Im(A_\pi S)\subseteq [\mathcal{H}]\_{\alpha}$, and implies (RC), we understand that the smaller $\alpha$ gets, the “closer” we are getting to (UC) being weaker than (RC).
> 3. We understand the difficulty, having in mind the recurrent message of the paper that metric distortion, and change of geometry between RKHS and L2 spaces in general, transmits. So, when writing this paper we have decided to explicitly use injection $S$ to clarify the change of spaces, and, for easier readability, omit putting each time the space subscript in the vector norms. However, we agree that Equation (10) merits extra clarity,  and we will clarify the meaning of the norms employed therein. We can also make a general comment that the $L^2_\pi(\mathcal{X})$-norm of a function $h\in\mathcal{H}$ is denoted as $\Vert Sh \Vert$.

---

> > ### Comment · Reviewer_GGn4 · 2023-08-11
> >
> > I thank the authors for addressing my questions the comments.
> > I keep my rating and recommendation.

---

### Official Review · Reviewer_1FxQ · 2023-07-26

**Soundness:** 4 excellent
**Presentation:** 4 excellent
**Contribution:** 4 excellent
**Rating:** 7
**Confidence:** 4

**Summary:**

This paper proves sharp upper bounds for eigenvalue estimation of a Koopman operator for a time-homogeneous Markovian dynamical system using either reduced rank regression or principal component regression. The bounds include operator norm error and metric distortion. These results are illustrated on simple models and some discussion of a higher dimensional molecular example is included. The error estimate also yields a design principle for kernels, based on spectral bias.

**Strengths:**

The paper is clearly written and articulates the both the theoretical results and their consequences on practical examples in a lucid manner. The results on metric distortion are novel for this problem. The paper synthesizes a number of existing arguments in a compelling way to generate a clear estimate on the spectral learning rate.

**Weaknesses:**

My impression is that the argument in Sec. 5 is not particularly new in the case that the HS norm error is used. But it is not very clear why to perfer the operator norm error.

**Questions:**

The estimator of metric distortion is introduced in the main text and some reference to its bearing on the experiments is made, but it is not clear from the experiments what role it plays. Can this be better explained?

If I understand correctly, the "ugly" kernel is chosen to make large the bias. Is there a way of making the deformation of the metric structure large in this example to illustrate the contribution of that term in the error?

In the appendix, it would be better to say in the alanine dipeptide example more explicitly how the RMSE is estimated. I assume the authors are forecasting the structure, but it could be some observable.

**Limitations:**

Yes, the conclusion captures limitations well. Negative impacts are not very relevant to this paper.

---

> ### Author Rebuttal · Authors · 2023-08-09
>
> We appreciate the reviewer's insightful evaluation and valuable comments. In what follows, we aim to address the highlighted weaknesses and respond to the reviewer's questions.
>
> __Weaknesses:__
>
> Indeed, the tools used to establish variance bounds in Section 5 are not particularly novel, and their fundamental concepts can be found in several papers on kernel-based learning, e.g., Ref. [18], [29]. The truly innovative aspects of our work in Section 5 are:
> 1. We derived the first operator norm error bounds, which usually pose greater challenges in achieving optimal rates. We have justified the necessity of using this norm in the global response to the reviewers.
> 2. In contrast to classical regression problems in reproducing kernel Hilbert spaces (RKHS) where the well-specified assumption implies that the regression function (conditional mean) lies in the RKHS, operator learning requires the much stronger assumption that the Koopman operator is a HS operator on the RKHS itself. This implies that both the mean and the law of the noise need to align with the RKHS, and that noise distribution needs to be sufficiently smooth. To bypass such strong assumptions we introduced a weaker regularity condition based on the cross-covariance operators that is applicable to a wider class of problems, e.g. practical systems like Langevin dynamics.
> 3. We establish that the RRR estimator, for which no error bounds were previously proven (neither in HS-norm), achieves the optimal learning rate.
>
> Finally, we remark that, while the state-of-the-art optimal rates presented by Li et al. (2022), Ref. [29], for HS-norm errors are provided under assumptions that do not guarantee the learning of a self-adjoint Koopman operator with a Gaussian kernel, our work demonstrates that the optimal operator norm rate is at least of the order $1/\sqrt{n}$.
>
> __Questions:__
>
> - The empirical estimator of the metric distortion plays a crucial role as the component of the empirical spectral bias $s_i$ in Theorem 4 (lines 291-29). This quantity is utilized in the experiments on Langevin dynamics to demonstrate its capability to predict eigenfunction estimation quality, as shown in Fig. 2, and for model selection in the Alanine-Dipeptide, as depicted in Fig. 3. We will provide further clarification on this matter in the revised version.
> - Indeed, you are correct. The choice of the "ugly" kernel is intended to amplify the operator norm error bias. However, since the family of kernels in this example is designed via Koopman eigenfunctions, we obtain that the eigenvalue decomposition of $C$ and $A_\pi$ differ only in eigenvalues and not eigenfunctions. Consequently, all three kernels induce no metric distortion on the estimator's eigenfunctions ($\eta(\psi_i)=1$).  So, to distort the metric, a different construction would be needed.
> - Indeed, we forecasted the 30 = 3*10 positions of the 10 atoms of the system. As you suggest, we will clarify this in the appendix.

---

> > ### Comment · Reviewer_1FxQ · 2023-08-14
> >
> >
> > Thanks for the clarification concerning the operator norm. I will keep my rating as accept.

---

### Author Rebuttal · Authors · 2023-08-09

We wish to thank all reviewers for their insightful evaluation of our paper. We appreciate all their comments and remarks, which we will incorporate in our revision. Before addressing each review in detail, we would like to point out some general remarks that apply to all of them.

__Assumptions & contributions:__ We thank the reviewers for the comments on the global assumptions of the paper. This made us notice that we should improve our discussion on self-adjointness and compactness in lines 67-69 and line 81, which will be rectified in the revision as follows:

Our main results come under different assumptions on the Koopman operator $A_\pi$:
- We prove _minimax optimal learning rates_ for operator norm error (Section 5). For these results to hold, $A_\pi$ just needs to be _bounded_.

- We derive _spectral learning rates_ (Section 6), further requiring $A_\pi$ to be _compact and self-adjoint_.

The additional assumptions in item 2 serve two purposes. Firstly, since there are currently no finite sample error bounds for eigenvalues and eigenfunctions of the Koopman operator, our results provide a first important step in this direction. Secondly, we emphasize that due to the peculiarities of the operator learning problem (in comparison to the simpler spectral estimation of matrices) even in the “well-conditioned” scenario of self-adjoint operators, bounding the operator norm error is insufficient to yield accurate estimates of the eigenvalues, leading to spurious phenomena.

__Operator norm instead of HS-norm:__ Koopman operators play a crucial role in science and engineering, offering valuable diagnostics and insights into dynamical systems' properties. Accurate estimation of their eigenvalues and eigenfunctions using the operator norm is essential for these applications, making spectral generalization bounds of paramount importance. However, existing theoretical guarantees with the Hilbert-Schmidt norm are often unsatisfactory, as they typically result in suboptimal bounds. In more detail, Section 5 is, in flavor, similar to existing recent work on optimal rates for Hilbert-Schmidt norm error in the estimation of the conditional expectation operator by Li et al. 2022, Ref. [29]. The main difference is that our bounds are derived for the operator norm (with a further novelty that two additional estimators are considered and under weaker assumptions). The significance of the operator norm stems from its critical property that, for every normal operator $A$ (even non-compact), $$dist(z,Sp(A)) :=\min_{\mu\in Sp(A)} | z-\mu | = \Vert (A-z)^{-1}\Vert^{-1}_{op}.$$ In contrast, using the Hilbert-Schmidt norm often leads to loose bounds for this distance. Employing the operator norm is crucial in obtaining sharp bounds for the spectral estimation of bounded operators and is a necessary step in addressing the problem of spurious eigenvalues, as is customary in spectral perturbation theory.

We will clarify the above points  in the revised version, providing better explanations of the limitations and highlighting essential directions for future research.

Once again we wish to thank the reviewers for their valuable feedback!
The Authors

---

### Decision · Program_Chairs · 2023-09-21

**Decision:**

Accept (spotlight)

**Comment:**

The authors tackle the important problem of learning nonlinear dynamical systems through learning the Koopman operator. This is an established approach but one that is lacking in theoretical guarantees. The authors give both strong/extensive non-asymptotic theoretical results (significant over previous work, with the first analysis of popular estimators), and provide accompanying experiments; the theory results seem to be informative for empirical phenomena (e.g., spurious eigenvalues). The paper received excellent scores, and the authors clarified with technical interpretations of the results. I recommend acceptance.